

# Capturing variation in floral shape: a virtual3D based morphospace for *Pelargonium*

Sara J. van de Kerke[1,2], Tiemen van Engelenhoven[1], Anne L. van Es[1], Laura Schat[1,4], Lisa M. van Son[1], Sverre Vink[1], Lia Hemerik[3], Robin van Velzen[1], M. Eric Schranz[1] and Freek T. Bakker[1]

[1] Biosystematics Group, Wageningen University and Research, Wageningen, Netherlands
[2] Hortus Botanicus Amsterdam, Amsterdam, Netherlands
[3] Biometris, Department of Mathematical and Statistical Methods, Wageningen University and Research, Wageningen, Netherlands
[4] Department of Ecology, Environment and Plant Sciences, Stockholm University, Stockholm, Sweden

Corresponding author
Sara J. van de Kerke,
saravandekerke@outlook.com

## ABSTRACT

**Background:** Variation in floral shapes has long fascinated biologists and its modelling enables testing of evolutionary hypotheses. Recent comparative studies that explore floral shape have largely ignored 3D floral shape. We propose quantifying floral shape by using geometric morphometrics on a virtual3D model reconstructed from 2D photographical data and demonstrate its performance in capturing shape variation.

**Methods:** This approach offers unique benefits to complement established imaging techniques (i) by enabling adequate coverage of the potential morphospace of large and diverse flowering-plant clades; (ii) by circumventing asynchronicity in anthesis of different floral parts; and (iii) by incorporating variation in copy number of floral organs within structures. We demonstrate our approach by analysing 90 florally-diverse species of the Southern African genus *Pelargonium* (Geraniaceae). We quantify *Pelargonium* floral shapes using 117 landmarks and show similarities in reconstructed morphospaces for nectar tube, corolla (2D datasets), and a combined virtual3D dataset.

**Results:** Our results indicate that *Pelargonium* species differ in floral shape, which can also vary extensively within a species. PCA results of the reconstructed virtual3D floral models are highly congruent with the separate 2D morphospaces, indicating it is an accurate, virtual, representation of floral shape. Through our approach, we find that adding the third dimension to the data is crucial to accurately interpret the manner of, as well as levels of, shape variation in flowers.

# INTRODUCTION

Variation in floral form continues to be an inspiration for a wide variety of research fields, ranging from taxonomy (*Linnaeus, 1758*), developmental biology (*Carr & Fenster, 1994*;

*Coen & Meyerowitz, 1991*; *Cubas, Vincent & Coen, 1999*; *Fenster et al., 1995*; *Luo et al., 1996*; *Mummenhoff et al., 2009*; *Parenicova et al., 2003*), evolution (*Darwin, 1877b*; *Reyes, Sauquet & Nadot, 2016*; *Sauquet et al., 2017*), adaptation, to pollination biology and speciation (*Darwin, 1877b*, *1877a*; *Fernández-Mazuecos et al., 2013*; *Gómez et al., 2016*; *Grant, 1949*; *Van der Niet & Johnson, 2012*). The term 'form' refers to a combination of size and shape (*Goodall, 1991*; *Zelditch, Swiderski & Sheets, 2012*). Whereas allometry is the study of the effect of size on the variation in morphological traits (*Klingenberg, 2016*), shape is defined as 'those geometrical attributes that remain unchanged when the figure is translated, rotated and scaled' (*Goodall, 1991*).

The total variation in shape occurring within a clade after scaling and aligning forms observed in its species is defined as the 'morphospace' (*Chartier et al., 2014*; *Foote, 1997*; *Gould, 1991*; *Wagner, 1996*), which can change depending on the taxa included in the study. Traditional versus geometric morphological methods (GMM) have been the subject of debate (*Adams, Rohlf & Slice, 2004*; *Rohlf & Marcus, 1993*). In GMM, landmarks placed on homologous structures capture the geometry of the studied object. Shape is maintained throughout the analyses, preserving the geometric relationships between structures (*Adams, Rohlf & Slice, 2004*; *Rohlf & Marcus, 1993*).

For any comparative study, measurement accuracy and precision are important to know about. A commonly used way to determine accuracy here is taxonomic sampling, assuming that more taxa included will yield better coverage. Often, larger clades are considered to be more informative because more taxa means more data. However, in a morphometric study, it might be more important to include a broad representation of the expected and potential morphological diversity in the sampling, irrespective of phylogenetic diversity since these do not necessarily go together.

When a floral GMM analysis is performed on a plant clade, maintaining the precision of gathering the data poses an additional challenge. Since plant morphology can be considered 'a process' (*Sattler, 1990*, *1996*), that is development, it is important to make sure that there is no noise from developmental signals in the data and its resulting morphospace, and hence that comparisons are made for the same ontogenetic stage across individual flowers. Ontogenetic noise can be prevented by deciding on a particular developmental stage for all individuals when measured. Full anthesis of the corolla is an example thereof (*Gómez et al., 2016*). However, studies have shown that different organs within individual flowers are not synchronised in their development (*Ronse De Craene, 2018*) and that species differ in the synchronisation of their floral parts (S.J. Van de Kerke, 2017, personal observations). Therefore, the floral parts of all individuals included in the study should be captured during the same ontogenetic stages, which poses a practical problem in data gathering.

Another practical challenge in floral GMM is the variation in copy number of included structures. For example, a species can display variation in number of stamens or petals within its flowers. This can be problematic because GMM studies are based on capturing homologous structures and therefore retaining accurate homology assessment is essential. Simply omitting copy number-variable structures from the analysis is not desirable since they represent evidence on shape. Assuming serial homology, and 'filling in' missing

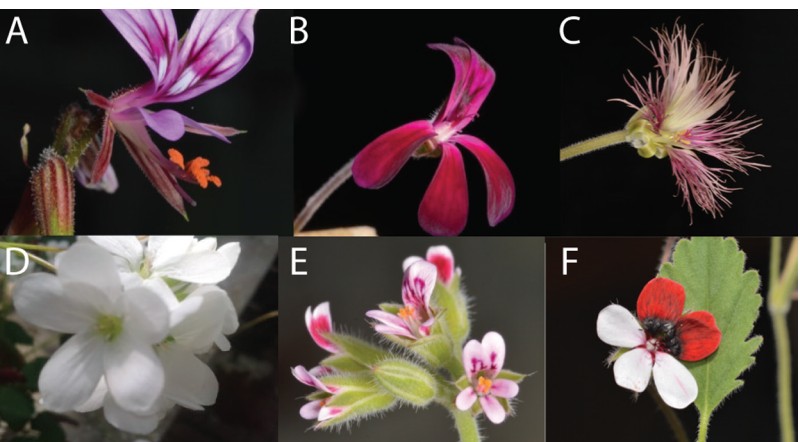

**Figure 1 Overview of variation in floral shape in _Pelargonium_.** (A) _P. caucalifolium_; (B) _P. sidoides_; (C) _P. caffrum_; (D) _P. cotyledonis_; (E) _P. columbinum_ and (F) _P. tricolour_. Pictures by F.T. Bakker and S.J. Van de Kerke.

copies could be one solution but the overall shape may be affected. Thus, handling such variation in morphs and their varying copy numbers is not straightforward.

We aim to address the GMM challenges outlined above, using the predominantly South African genus _Pelargonium_ (Geraniaceae) as a model. The genus is known for its stunning floral and vegetative diversity across its ~280 species (_Bakker et al., 2005_, _1999_, _Jones et al., 2009_; _Jones, Cardon & Czaja, 2003_; _Nicotra et al., 2008_; _Röschenbleck et al., 2014_; _Struck, 1997_; Fig. 1) and has been the subject of wide-spread breeding and horticulture (_Becher et al., 2000_; _James, 2002_; _Miller, 2002_). Roughly 70% of the genus occurs in the South African Greater Cape Floristic Region (GCFR; _Linder, 2003_; _Manning & Goldblatt, 2012_; _Snijman, 2013_), other species occur in eastern Africa, Namibia, Asia Minor, the Arabian peninsula, Madagascar and Australia (_Bakker et al., 2005_). Phylogenetic relationships within the genus are well known (_Bakker et al., 2005_; _Röschenbleck et al., 2014_; _Van de Kerke et al., 2019_) and show a pattern of deep splits as well as more recent species radiations (i.e. the geophytic sect. _Hoarea_).

_Pelargonium_ flowers are specialised when compared with the remainder of the Geraniaceae clade (i.e. _Geranium_, _Erodium_, _Monsonia_ and _California_), as they exhibit strongly zygomorphic corollas and possess nectar tubes that are formed from the receptacle (_Albers & Van der Walt, 2007_; _Bakker et al., 2005_; _Goldblatt, Manning & Bond, 2000_; _Hodges, 1997_; _Hodges & Arnold, 1995_; _Manning & Goldblatt, 2012_; _Tsai et al., 2018_; _Van der Walt & Vorster, 1981_, _1988_), not adnate to the pedicel, which is unique in angiosperms (_Hodges, 1997_; _Tsai et al., 2018_). Throughout _Pelargonium_, variation in floral shape occurs in a number of ways. Most strikingly, the orientation of the petals ranges from highly zygomorphic (_P. fulgidum_) to almost actinomorphic (_P. cotyledonis_). Secondly, the variation in petal copy number occurs between and within a species (i.e. _P. caucalifolium_) and alters between five (the 'standard' in Geraniaceae), four (_P. tetragonum_), two (_P. dipetalum_), and can even be missing (_P. apetalum_). Third, the shape of the petals varies tremendously: from slender and elongated (_P. paniculatum_) to almost round (_P. inquinans_). _Pelargonium_ exhibits a range of pollination syndromes,

including species of long-tongued hovering flies (*Tabanideae*, *Bombyliidae*, and *Nemestrinidae*), bees (*Apidae*, *Anthophoridae*, *Megachilidae*), wasps (*Vespidae*), and beetles (*Scarabaeidae*; *Struck, 1997*). Some syndromes are highly-specialised, as in the geophytic *P. appendiculatum*, with a limited distribution range in the Strandveld along the South African west coast (*Marais, 1999*), which has a nectar tube of 10 cm long, while no pollinator with a suitable proboscid is known. Or *P. fulgidum* from the coastal Fynbos, with its highly zygomorph and probably bird-pollinated flowers, comparable to the E African *P. boranense*. Another extreme, but rather generalist, example is the oceanic island endemic *P. cotyledonis* (occurring on St. Helena) where the nectar tube is reduced to <1.5 mm, and where this species possibly reverted to a generalist (or symplesiomorphic) state including a actinomorphic corolla, after arrival on the relatively insect-poor and remote island. Nectar tube length in *Pelargonium* appears to be a driver of speciation rate, whereby speciation rate seems to decrease with an increase in nectar tube length and is associated with small clade size (*Ringelberg, 2012*). The wide variety of known pollinators for *Pelargonium* is reflected in nectar tube length, whereby it matches the proboscis of the pollinator species. The extent to which the pedicel length compares with the nectar tube differs greatly among species (*Bakker et al., 2005*; *Manning & Goldblatt, 2012*; *Tsai et al., 2018*). This could indicate pedicel length evolution is independent from nectar tube length, but whether it is a potential constraint on nectar tube length change is not clear.

In this study, we infer the floral morphospace for the corolla and the nectar tube across *Pelargonium*, which is an efficient way to visualise the amount of shape variation within and between species, and will help addressing the GMM issues outlined above. We use two-dimensional (2D) photographs to form virtual three-dimensional (virtual3D) representations of flowers in order to quantify floral shape in 90 *Pelargonium* species. We explore the diversity of floral forms within the genus and using this dataset as a case study we apply GMM methods to determine and compare natural variation in floral shape.

## MATERIALS AND METHODS

### Flower data sampling

Floral shape was compared for 90 *Pelargonium* species growing in living collections in The Netherlands, Germany and in South Africa (see Table S1 for an overview of species, numbers of individuals, and location). The sampling covers approximately 32% of known species in the genus and includes 436 individual flowers. We covered the potential morphospace as adequately as possible based on known extreme floral forms from taxonomic studies (*Albers et al., 1995*; *Van der Walt, 1985*; *Van der Walt & Boucher, 1986*; *Van der Walt & Van Zyl (nee Hugo), 1988*), but not-necessarily representing phylogenetic diversity.

### Geometric morphometric data collection

We separately selected flowers with corollas in anthesis and other flowers with mature stamens (used as proxy for maturity of the nectar tube, which was confirmed by eye) to limit possible ontogenetic effects on measured shape. We digitally photographed each

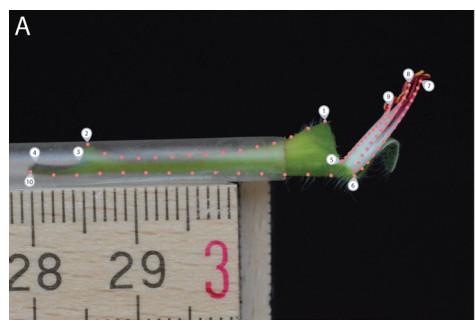
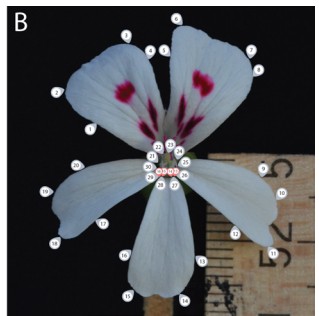

**Figure 2 Landmark placement for the TUBE (A) and PETAL (B) datasets.** We defined a set of 10 landmarks covering the overall outline and main features of the nectar tube and 75 sliding landmarks tracking its curvature, as well as that of the shortest, longest and an average length stamen (in sets of 15, equally distributed over the stamen curve). For the PETAL dataset 30 landmarks were placed along the outline of the corolla and the opening of the nectar tube using primary and secondary veins and petal attachment as a guide (grey labels). For specimens with four petals, we assumed that for the middle anterior petal the meristem is present but does not develop and landmarks allocated for this petal were placed with zero length from the petal base (pink labels).

flower using a standardised procedure in front and side view to avoid positional effects on measured shape. For each photograph, we defined a set of landmarks to provide comprehensive coverage of the specimen. We used both primary landmarks on homologous positions as well as sliding landmarks along a curve between two fixed primary landmarks. A datafile was created using tpsUtil (*Rohlf, 2015*) and landmarks were placed using tpsDig v. 232 (*Rohlf, 2010*).

For the side view photograph, covering the nectar tube aspect, we defined a set of 10 landmarks covering the overall outline and main features of the nectar tube and 75 sliding landmarks tracking its curvature, as well as that of the shortest, longest and an average length stamen (in sets of 15, equally distributed over the stamen curve; Fig. 2A, grey labels). We labelled this data set TUBE (containing 134 individuals, Table S1).

For the front view photograph, we followed the landmarks per petal as defined by *Gómez, Perfectti & Camacho (2006)* and placed 30 landmarks along the outline of the corolla of the five petals and the opening of the nectar tube using primary and secondary veins and petal attachment as a guide (Fig. 2B, grey labels); midrib veins appear to be absent in *Pelargonium* petals. We labelled this data set PETAL (containing 287 individuals, Table S1). For specimens with four petals, we assumed that for the middle anterior petal the meristem is present but does not develop (*Ronse De Craene, 2018*). Therefore, landmarks allocated for this petal were placed but with zero length from the missing petal base (Fig. 2B, pink labels). A 5 mm scale bar was included in each picture to be able to represent all landmark coordinates on the same interval scale.

## Creating virtual3D representations from two 2D photographs

To be able to understand how shape variation happens at the level of the complete flower we linked individuals from both datasets at the species level. One-on-one pairing of individuals in the separate TUBE and PETAL databases was not possible because the flowers we used are not the same for both datasets (as a result of the separate sampling),

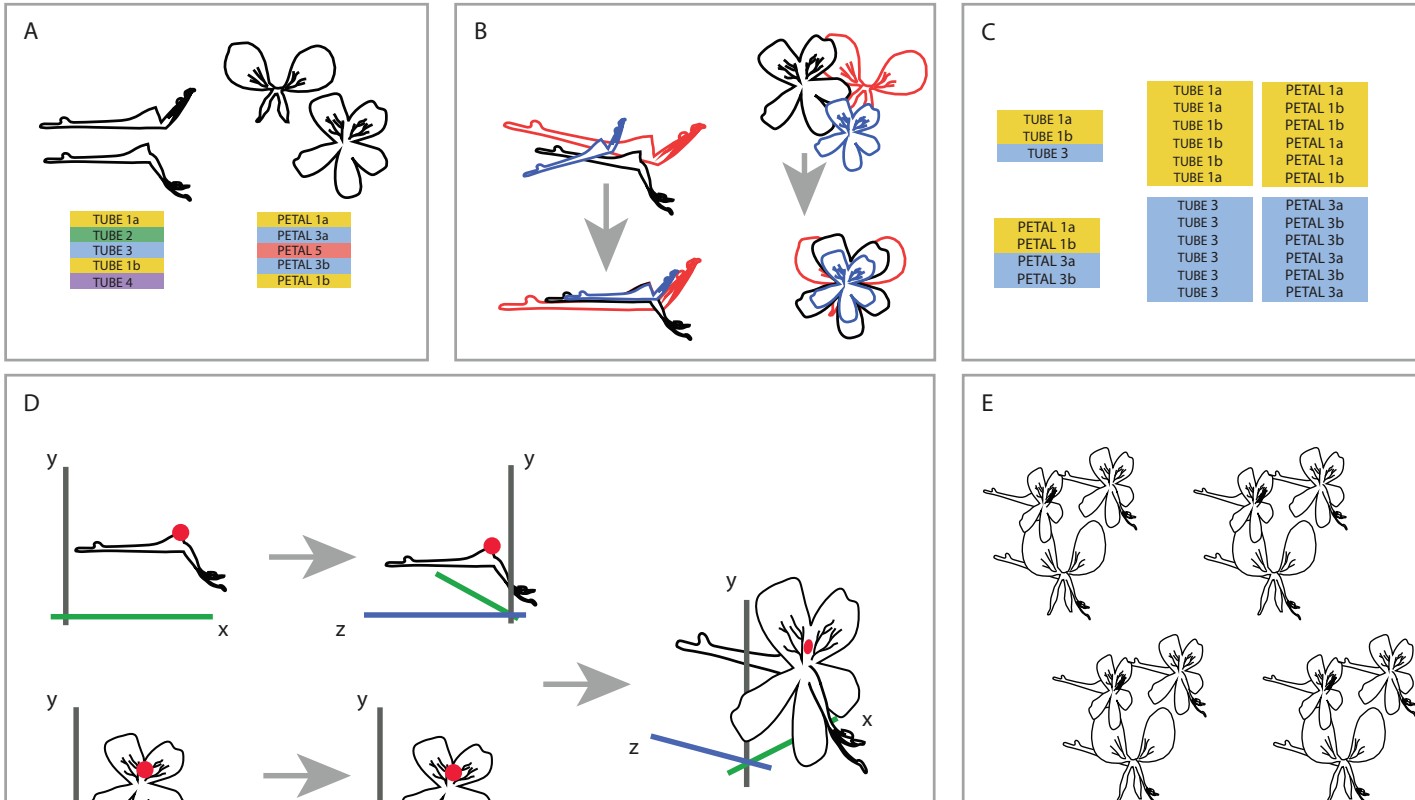

**Figure 3 Creating one 3D virtual flower from two 2D photographs.** (A) The two separate datasets (TUBE and PETAL), with limited overlap between and within species. (B) Generalised Procrustes Analysis is performed on the TUBE and PETAL datasets separately in order to filter out all non-shape variation. Size component is then reintegrated by multiplying each individual with its calculated centroid size. In this way, all specimens are aligned based on their landmarks, without removing size information. (C) Species present in both TUBE and PETAL datasets are selected. In order to link species in both data sets, a random individual from dataset TUBE is then drawn for the first species and combined with a random individual of the same species from dataset PETAL. This was done six times per species, with replacement. (D) To integrate the two 2D datasets into a single 3D dataset, a common anchor point is defined in both the TUBE and the PETAL datasets, corresponding here with the top of the opening of the tube. A third coordinate is then added to the coordinate data, effectively making it 3D. (E) This process is repeated for all individuals in the set selected in the linking step. See text for further details.

nor were individuals sampled from the same plant. Therefore, we designed a random sampling bootstrapping method based on the TUBE and PETAL datasets (see Fig. 3).

First, we reoriented all individuals in both TUBE and PETAL dataset in the same position before we connected them to assure a virtual 3D flower that is congruent with actual morphology. To that extent, we performed an initial Generalized Procrustes Analysis (GPA) on the TUBE and PETAL datasets separately in order to align specimens and remove size components. Subsequently, we reintegrated the size component in order to retain actual size of the individual when coupling them from TUBE and PETAL datasets. This was accomplished by multiplying each individual with its calculated centroid size. In this way, we orientated all specimens in the same position based on their landmarks, without removing size information (Fig. 3B).

Next, we selected the species present in both TUBE and PETAL datasets. For each species, the number of individuals in each dataset was counted and we recorded at which row in the dataset a new species starts. In a linking step, a random individual from a certain species in dataset TUBE was then drawn and combined with a random individual of the same species from dataset PETAL. This was done six times per species, with replacement (Fig. 3C).

To integrate the two 2D datasets into a single virtual3D dataset, a common anchor point was defined in both the TUBE and the PETAL datasets, which we chose to correspond to the top of the opening of the nectar tube. In the TUBE dataset, the first landmark was chosen as anchor and for PETAL we defined the anchor to be the average of landmarks 22 and 23, as these anchors are homologous (Fig. 3D).

A third coordinate was then added to the two 2D coordinate datasets (PETAL and TUBE), effectively making a virtual3D image (dataset VIRTUAL3D). For the PETAL data set we kept the original $x$ and $y$ values and add a $z = 0$ coordinate to all landmarks. In this way, we 'forced' the corolla of the flower to be flat because we do not have data on the curvature of the petals. For TUBE the coordinate system was altered from $x, y$ to $z, y$, which effectively becomes the depth of the flower. This alteration is relative to the coordinate combination of the anchor point defined previously, landmark TUBE 1 and landmarks PETAL 22-23, that is they are placed perpendicular to each other, around the anchor. Therefore the coordinates became negative for the nectar tube and positive for the stamens. The value $x = 0$ was added for all TUBE landmarks, again resulting in a forced flat object. The new $x$ and $y$ TUBE coordinates were then transposed relative to the landmarks 22–23 anchor point of PETAL (Fig. 3D), with which they were subsequently combined. The new virtual3D coordinates were written to a .txt file suitable for later analysis with Geomorph and other GMM packages. This process was repeated for all combinations of individuals in the set selected in the linking step described above (Fig. 3E).

This process was repeated 20 times to assess the structure in the virtual3D flower data, and hence its stability, resulting in 20 bootstrap pseudoreplicate datasets containing $6 \times 68$ species = 408 virtual flowers each, which we label VIRTUAL3D$_i$ (with $i$ =1,…,20). We combined all resulting 8160 virtual flowers in VIRTUAL3D, a dataset which we use for further analyses.

## Morphometric analysis

Landmark coordinates in the TUBE, PETAL, VIRTUAL3D, and all VIRTUAL3D$_i$ datasets were each aligned using a final GPA, extracting the shape information (*Rohlf & Slice, 1990*). Results were projected into tangent space to summarise and explore actual (TUBE, PETAL) and virtual (VIRTUAL3D) floral shape variation across *Pelargonium* species. Shape changes associated with principal components where illustrated using thin-plate spline deformation plots.

Part of the shape variation observed in petals could result from corolla asymmetry. We assessed this aspect in *Pelargonium* corolla shape using the function 'bilat.symmetry' in Geomorph. We performed a Principal Component Analysis (PCA) on the symmetric component of the resulting dataset (PETAL.sym). Results were highly similar with the

PCA results of the original PETAL datasets (not shown). We therefore did not include this aspect in further analyses.

We conducted a PCA on the GPA-aligned coordinates for each of the VIRTUAL3Di datasets. PCA results for the 20 VIRTUAL3Di datasets are highly congruent (not shown, data will be made available). This indicates that there is high consistency in our data and that bootstrap subsampling seems justified for connecting the differently samples TUBE and PETAL datasets. We therefore decided to continue our analyses with the VIRTUAL3D dataset including all 8160 virtual flowers, as this dataset assures an even coverage of all included species and we consider it the most inclusive.

We assessed measurement error by digitising a random subsample of ten individuals twice, several months apart. Differences between the sets where measured using Procrustes ANOVA following *Savriama (2018)* in MorphoJ (*Klingenberg, 2011*). We found inter-individual variation to be highly significant ($P < 0.0001$) and the effect of digitisation on both shape and size of no importance (not shown).

Nectar tubes occur inside the pedicel in *Pelargonium* species and pedicels can be 'occupied' by nectar tubes to varying degrees. As this may in fact present limits to nectar tube length it could constrain nectar tube evolution and be relevant to floral shape exploration. We therefore decided to extract the aspect of 'occupancy' (i.e. the pedicel plus receptacle length relative to receptacle length) from our data in the following way: for each individual in the TUBE dataset, we extracted relative nectar tube and pedicel length from the TUBE dataset using the function 'interlmkdist'. We calculated the ratio between the nectar tube and pedicel length and visualised it in the TUBE PCA plot, indicated by the transparency of the individual mark.

All analyses were performed in R v.3.2.2 (*R Core Team, 2015*) using the Geomorph library v.3.0 (*Adams & Otárola-Castillo, 2013*). All R scripts can be found in R scripts 1, 2, and 3.

## RESULTS

Our analysis on 68 *Pelargonium* species identified a wide variety of floral shapes across and within the species examined (see Fig. 1). Figure S1 shows the mean consensus configuration and Procrustes residuals (i.e. differences between observed and estimated value) calculated for the TUBE and PETAL datasets using the GPA. The figure illustrates the variability in landmarks around the calculated mean shape (in blue). What is striking is that halfway through the nectar tube we see a constrained area where variation is limited compared to the base of the pedicel (Fig. S5A). In addition, in Fig. S5B it is conspicuous that the anterior petals are more restricted in shape variation than the posterior petals.

We conducted a Principal Component Analysis (PCA) on the GPA aligned coordinates for each of the TUBE, PETAL, and VIRTUAL3D datasets in order to assess variation in shape. For the TUBE dataset, the first PC accounts for 47% of the total variation present across the species and the first four axes explain more than 90% of the data (Fig. 4A; Fig. S2). The first two PCs and corresponding shape outlines of the extremes are plotted in Figs. 4A and 4B, respectively. The variation in shape explained by the first PC corresponds
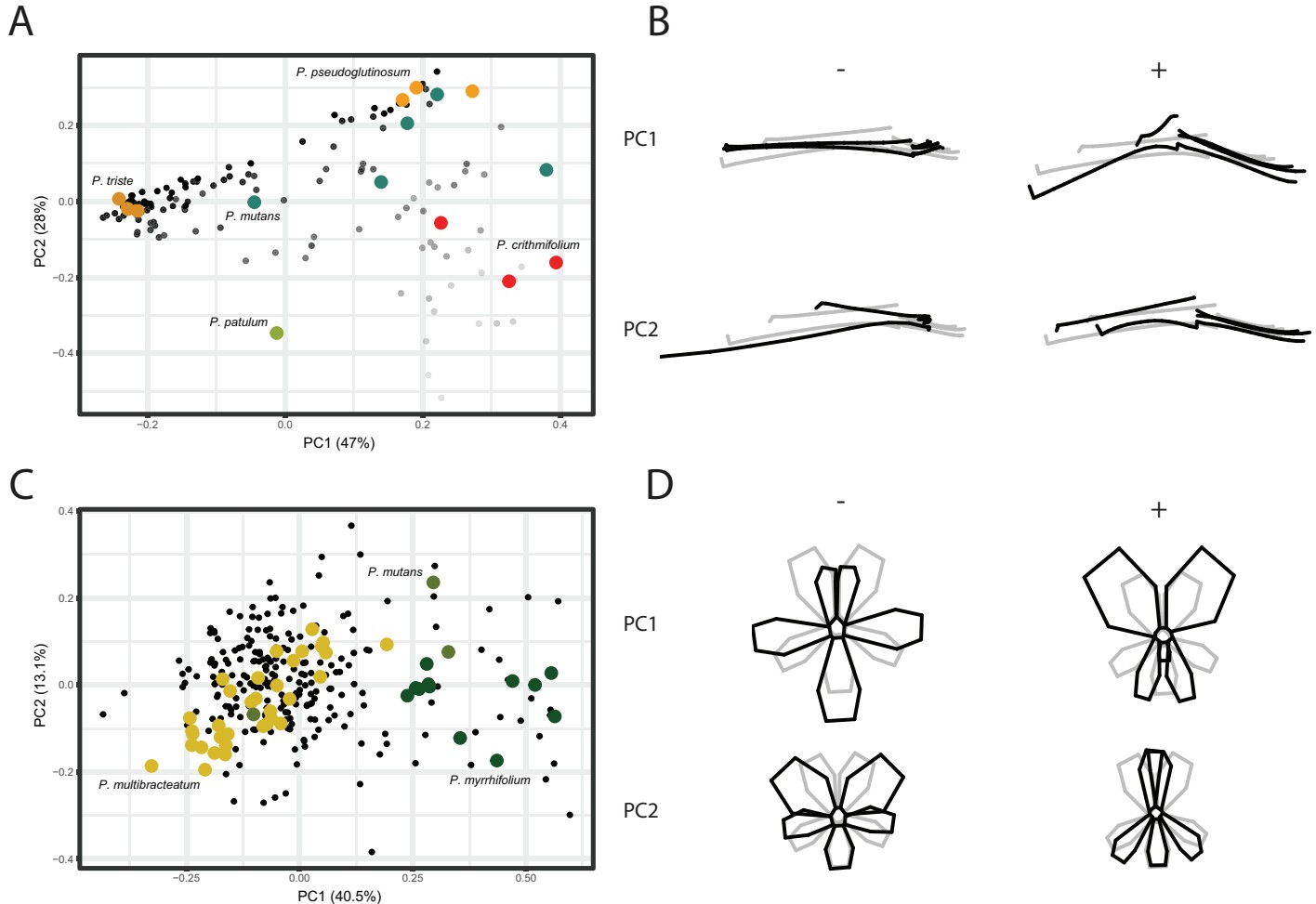

**Figure 4 PCA analysis on TUBE and PETAL datasets.** (A) PC1 and PC2 of PCA on TUBE dataset. Colours correspond with selected species: *P. triste* (brown), *P. mutans* (blue), *P. patulum* (green), *P. crithmifolium* (red) and *P. pseudoglutinosum* (orange). (B) Shape outlines corresponding to extremes on axes for PC1 and PC2 of TUBE dataset showing calculated mean shape (grey) and warped extreme shape (black). (C) PC1 and PC2 of PCA on PETAL dataset. Colours correspond with selected species: *P. multibracteatum* (yellow), *P. myrrhifolium* (dark green) and *P. mutans* (green). (D) Shape outlines corresponding to extremes on axes for PC1 and PC2 of PETAL dataset showing calculated mean shape (grey) and warped extreme shape (black).

to the length of the nectar tube relative to the pedicel. On the negative extreme of the axis, nectar tubes are elongated and have the same length as the pedicel. On the positive extreme, nectar tubes are much shorter than the length of the pedicel and, in addition, the opening of the nectar tube is wide. PC2 corresponds to the curve of the stamens. Individuals on the negative extreme of the PC have stamens that are so curved they are doubled up on themselves, while those on the positive side have elongated stamens (Fig. 4B, PC2). For some species multiple accessions are included, and in Fig. 4A we find them spread in varying degrees across the morphospace. Examples are *P. mutans* (in green) and *P. crithmifolium* (in red) along both PC1 and PC2. Other species appear to be much more clustered, such as *P. triste* (in brown) and *P. pseudoglutinosum* (in orange). Overall, a clear pattern emerges of individuals distributed along a trajectory corresponding with the ratio between the length of the nectar tube and pedicel towards a boundary

reflecting a physical barrier (indicated by transparency of the markers in Fig. 4A). Individuals with a low nectar tube-pedicel ratio occupy the lower region of the PC plot while individuals towards the top have an increasingly higher nectar tube-pedicel ratio, that is having nearly the same length. This boundary is also reflected in the 'avoided area' in the *Pelargonium* TUBE morphospace just above it. In this area, the nectar tube of a hypothetical flower would be longer than the pedicel of that individual, and this is not possible for *Pelargonium* flowers as nectar tubes are not free but deep receptacular nectaries (*Tsai et al., 2018*).

Compared to the results of the TUBE dataset, the PCA results of the PETAL dataset are more centralised. In Figs. 4C and 4D, the first two PCs and shape outlines are plotted. The first PC (explaining 40% variation, Fig. S2) corresponds to the position and number of petals in the flower. On the negative extreme of the axis, flowers consist of five petals with the two posterior ones close together and the three anterior petals spread out. On the positive extreme, the two posterior petals are enlarged and only two anterior petals appear to be present. PC2 (13%) corresponds to the distribution of the petals over the corolla. On the positive extreme of the PC, the posterior petals are narrow and overlap, while on the negative side the posterior petals are rounded. Overall, individuals cluster around the mean shape (as *P. multibracteatum* (yellow)) while other species show within-species variation with individuals that spread toward the positive extreme of PC1 (*P. myrrhifolium* (dark green)). A few species (as *P. mutans* (green)), with high within-species variation, are found across the entire PCA spectrum.

For the VIRTUAL3D dataset, containing 8160 virtual flowers, the first PC accounts for 41% of the total variation present across species, with the first 5 axes collectively explaining >80% of the data (Figs. 5A and 5B; Fig. S2). Shape outlines illustrating the extreme forms are shown in Fig. 5C. The variation in shape explained by the first PC corresponds to a zygomorphic flower, with corolla size varying with regards to pedicel length. On the positive extreme, individuals have a short pedicel and nectar tube and a large corolla while flowers on the negative extreme show a more elongated nectar tube and a relatively small corolla. Individuals from all species are spread along this axis, showing a high variability in nectar tube and pedicel elongation and no clustering. PC2 (19%) corresponds to the length and curvature in stamens, with virtual flowers on the negative extreme showing straight stamens and those on the positive extreme showing highly curved ones. More importantly, this PC appears to correspond with the occupancy of the pedicel by the nectar tube, whereby we either see a long pedicel and relatively short nectar tube (positive side) or a nectar tube that 'spills over' the entire pedicel (negative side). Individuals from all species are spread along the axis but with an emphasis toward the negative extreme, suggesting a trend towards individuals with a high filling ratio. PC3 (14%) again (as PC1) appears to correspond with the occupancy of the pedicel by the nectar tube as well as the length and orientation of the stamens. In individuals toward the positive end of this axis, the nectar tube completely occupies the pedicel and stamens are stretched out. On the negative side, only a small part of the pedicel is taken up by the nectar tube and stamens are small. No clustering is observed and individuals are spread along the axis but with a strong emphasis on the negative end of the spectrum. Individuals within

species are spread in varying degrees around the morphospace, such as *P. mutans* (in green) along PC1, PC2, and PC3. Other species vary along a number of PC axes, as *P. crithmifolium* (in dark blue) is variable along PC1 and PC3, but not along PC2. Lastly, some species are overall much more clustered, such as *P. pseudoglutinosum* (in orange).

## DISCUSSION

In this study, we explore the potential of combining two 2D photograph-based datasets of floral morphology into a single virtual3D flower giving us the opportunity to bring together multiple layers of shape variation. Using this method, we are able to investigate the tremendous floral diversity of *Pelargonium* species using 3D geometric morphometrics based on the combined nectar tube and corolla perspective. Our virtual3D dataset gives a more nuanced view on shape variation in *Pelargonium* than the separate TUBE and PETAL perspectives, as we find the corolla perspective to be of less importance (see below). Our approach, although virtual3D, can serve as a low-cost alternative to emerging high-tech robotic and photogrammetry-based approaches to 3D geometric morphometrics.

### Geometric morphometrics

*Pelargonium* flowers exhibit high variability in their floral shape with species ranging between zygomorphic to near-actinomorphic corolla shape (*P. cotyledonis*), varying in petal copy number (between five (most common in Geraniaceae), four (i.e. *P. caucalifolium*)), two (in *P. dipetalum*; not included) and zero (in *P. apetalum*; not included), and with lengths of nectar nectar tubes varying between zero to ten cm (*P. appendiculatum*; not included). The variation in floral shape present in the VIRTUAL3D dataset as depicted in Fig. 5 corresponds to this known variation in *Pelargonium* flowers, as well as with the separate PETAL and TUBE datasets (Figs. 3 and 4). Findings of the separate PETAL and TUBE datasets have now been put into perspective, giving us a better understanding of which changes in *Pelargonium* floral shape are relevant.

Resembling the results of the TUBE dataset, the elongation of the nectar tube and size of the corolla are the most variable traits among the species included in the VIRTUAL3D morphospace (PC1, 41%). This trait corresponds to the unique nectar tube pollinator syndrome featured in *Pelargonium* and correlates with its variable palette of pollinators and pollination syndromes (*Struck, 1994*, *1997*). We know in some species the nectar tube to be almost completely missing (Fig. 5C, as for example in the oceanic island endemic *P. cotyledonis*, probably pollinated by bees) or in *P. hirtum* with 3 mm short nectar tubes. The latter is closely related to *P. appendiculatum* (probably pollinated by now-extinct long-tongued hovering flies) where the nectar tube is elongated to almost ten cm length (*Struck, 1997*).

Corresponding to PC2 (19%), and linked to inferred shifts in pollinators, is the curvature of the stamens. Along the PC, we find a shift of stamen shape ranging from short and straight to long and curved. For some hovering pollinator species, the stamens are thought to 'move out of the way' of the nectar tube entrance by means of a large curve in

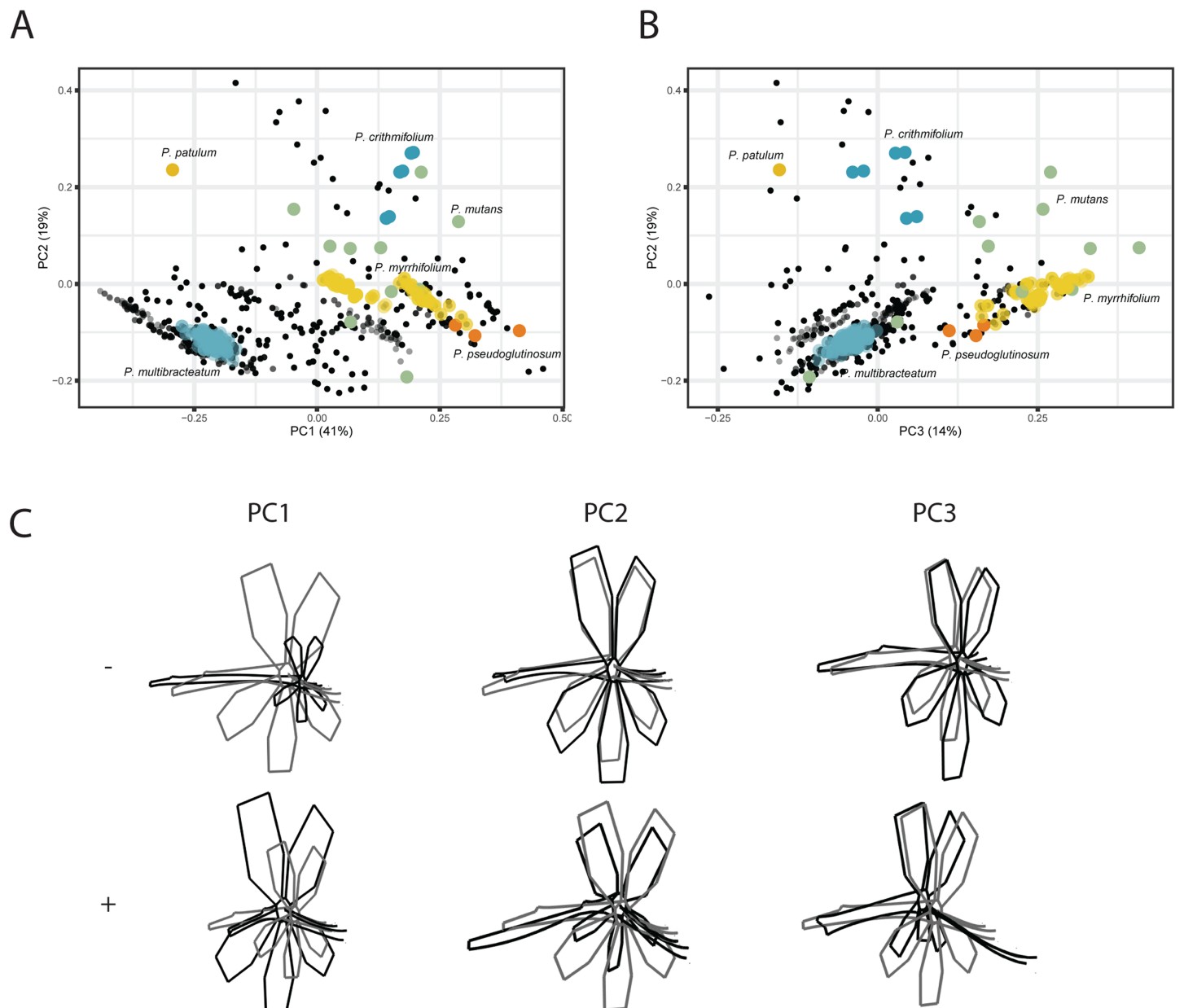

**Figure 5 PCA analysis on VIRTUAL3D datasets.** (A) PC1 and PC2 and (B) PC3 and PC2. Colours correspond with selected species: *P. multibracteatum* (light blue) *P. triste* (brown), *P. mutans* (grey), *P. myrrhifolium* (yellow), *P. patulum* (brown), *P. crithmifolium* (dark blue), and *P. pseudoglutinosum* (orange). Intensity of colours indicated number of individuals stacked. (C) Shape outlines corresponding to extremes on axes for PC1, PC2 and PC3 of VIRTUAL3D dataset showing calculated mean shape (grey) and warped extreme shape (black).

the filament, both increasing accessibility to the flower (*Goldblatt & Manning, 1999*; *Manning & Goldblatt, 1996*) and enhancing contact of anthers and insect abdomen and head (*Goldblatt & Manning, 1999*). This would correspond to the long and curved stamens of *Pelargonium* species pollinated by long-tongued, hovering insects such as species from the *Tabanideae*, *Bombyliidae*, and *Nemestrinidae* (*Struck, 1997*). The short and straight stamens on the other end of the spectrum would then correspond with the

association with short-proboscid, landing pollinator species, such as *Anthophoridae*, *Megachilidae*, and *Vespidae* to increase potential pollen transfer.

The occupancy of the pedicel by the nectar tube, which corresponds to both the second as well as the third PC (11%) in the VIRTUAL3D as well as the TUBE dataset is a relatively unexplored trait in *Pelargonium* literature. Recent studies found nectar tube length to be dependent on both rate of cell division and duration of nectar tube growth (*Tsai et al., 2018*). As the authors indicate, these mechanisms do not fully account for differences in nectar tube length, suggesting other evolutionary influences. *Ringelberg (2012)* found nectar tube length to be significantly correlated with speciation rate, whereby speciation rate appeared to decrease with increased nectar tube length. To what extent the physical boundary of pedicel length for tube length change would play a role in speciation remains to be investigated.

The distribution of virtual flowers over the first three PCs of the VIRTUAL3D morphospace varies and appears to be the results of interaction between the TUBE and PETAL morphospaces. In the TUBE morphospace, we see a clear boundary limiting the distribution of individuals based on the ratio of nectar tube and pedicel length (Fig. 4A). In the PETAL morphospace on the other hand, the majority of species cluster together around the mean shape, indicating that there is variation to a limited extend. Some species in the VIRTUAL3D morphospace are highly variable and occur throughout large areas of the morphospace (for instance *P. mutans* (grey)) while others occupy a much smaller area (e.g. *P. multibracteatum* (light blue)). The former pattern does not directly correspond with a high individual count in PETAL and TUBE datasets. Certainly, in cases as *P. crispum* the low variability is the consequence of there being only one individual in the PETAL and TUBE datasets. As a result, over all the bootstrap iterations, only a single virtual-flower is included in the final analysis. But in other cases, as for instance with *P. multibracteatum*, multiple individuals are included in the separate datasets and still we find a narrow distribution in the morphospace. This would indicate that these species are canalised in their floral shape development, possibly having implications for their pollinator dependence.

Surprisingly, the results of the VIRTUAL3D dataset as discussed above are highly congruent with the results of the TUBE dataset while the PETAL dataset does not appear to have much influence since we do not find the variation in shape along PC1 in the PETAL dataset (variability in length of the fifth petal) until the third PC (14%). Rather the size of the corolla relative to the length of the nectar tube is found to be of more influence in the VIRTUAL3D dataset. The variability in nectar tube and stamens, combined with this relative size difference of the corolla, thus seems to be more relevant for distinguishing different shapes and presumably for attraction of pollinators.

## virtual3D connexion of 2D data sets

The combination of separate 2D datasets into a single virtual3D dataset by creating virtual flowers as we demonstrate here complements existing 3D approaches (*Van der Niet et al., 2010*). We find the main PCs of the VIRTUAL3D dataset summarise the variability in shapes as presented in the separate TUBE and PETAL datasets and therefore consider
them to accurately portray the natural variation found in *Pelargonium* flowers (based on visual inspection). Having rendered the flower in virtual3D, we can now investigate the interaction between floral parts in more detail.

Our method enables us to circumvent a main issue in morphometric studies on flowers and thus to increase the precision of the data: asynchronicity in anthesis of floral parts. The moment of anthesis of floral parts differs both between and within species (commonly referred to as 'parcellation'). This makes it impossible to pinpoint an ontogenetic stage for the entire flower that is the same for all species. We argue that anthesis is the most relevant ontogenetic stage for reproduction as well as pollinator attraction and thus is the most meaningful stage to include in our study. Following other plant studies (*Berger et al., 2017*; *Gómez, Perfectti & Camacho, 2006*; *Gomez, Perfectti & Klingenberg, 2014*; *Savriama et al., 2012*), we decided to include all floral parts at their own, separate, anthesis. This results in the separate datasets of the TUBE (containing the nectar tube and stamens) and the PETAL (containing the corolla). We consider the combination of nectar tube and stamen floral parts in the TUBE dataset plausible since we suspect the flower's reward system to develop approximately in concert with the contact apparatus, in order to 'fit' the visiting pollinator.

A drawback of combining the different floral parts each at their own anthesis is that we construct 'virtual-flowers' (i.e. not actually occurring) from our data. As a result, the morphospace is arguably not biologically and temporally accurate. However, we argue that gathering the data in the same, homologous, ontogenetic stage gives us the advantage of not polluting our data with unwanted developmental signal and enables the testing of evolutionary hypotheses regarding dynamic (un)coupling of compartments (S.J. Van de Kerke et al., 2017, personal observations).

Another problematic issue in plant geometric morphometrics is the variability in copy number within floral parts. A striking example of this phenomenon in *Pelargonium* is the variability in petal number, varying to two and four from the symplesiomorphic number of five (*Röschenbleck et al., 2014*). This variability makes it seemingly impossible to include all intended landmarks since they have to be placed on homologous structures. Not including these landmarks in the study is not desirable as they do represent an important difference in shape between species. Likewise, it is not an option to treat them as 'missing' or 'NA' since the flower did not drop the petal by accident, but it is simply not present. Ideally, we would like to confirm the presence of petal primordia in an electron microscopy study. Based on literature describing the occasional loss of petals (*Ronse De Craene, 2015*, *2018*), we choose to simulate 'missing' petals as if it is present, but with a length of zero (Fig. 2). We consider the influence of this simulation on morphospace results as limited since the variation between four and five petals is only visible on the fifth PC (4%) of the VIRTUAL3D dataset. We admit this approach is conceptually problematic because we assume the petal to be present, but operationally warranted because we find no effect in our resulting morphospaces.

Unfortunately, we were not able to achieve complete matching in taxonomic coverage between the separate TUBE and PETAL datasets because sometimes there were no flowers in anthesis available for both datasets. The separate morphospaces therefore have a

higher taxonomic sampling than the VIRTUAL3D dataset (68 for the VIRTUAL3D compared to 82 in TUBE and 90 in PETAL). This is an insurmountable drawback in combining the datasets, since in morphometric studies all landmarks need to be present in all included specimens in currently available software packages. Estimating missing landmarks, as can be performed using Geomorph, is not desirable when a large part of the studied shape of an entire species is missing. In such case the average *Pelargonium* shape is superimposed on a set of individuals and their unique shape is lost.

More important than high taxonomic coverage in the VIRTUAL3D dataset is to ensure accuracy of the data by adequate coverage of morphological extremes in the morphospace, which is not driven by the number of species included but by the shapes. In the case of *Pelargonium*, we have several 'missing' shapes that we were not able to include in the sampling (we did not encounter them while flowering) that will probably change the morphospace were they to be included. For example, we did not have the opportunity to include species such as *P. endlicherianum* and *P. dipetalum*, that only have two posterior petals. Likewise, we could not include species showing highly reflexed petals (for example *P. luridum*) as well as the peculiar, keel-flowered shaped *P. rapaceum* and the allopolyploid *P. quercetorum*. Notwithstanding these gaps in the prospective morphospace, we are confident we reconstructed a fair representation of overall variability in floral shape found in *Pelargonium* and therefore provide a solid basis for exploring floral shape in this clade.

## CONCLUSIONS

This study provides a new approach for geometric morphometric analysis of floral shape in virtual3D. Our method uses a semi-automated approach to combine 2D shape data of various data sets to include multiple morphological modules. It offers unique benefits to complement established imaging techniques by (i) providing a bootstrapping method to help acquiring adequate coverage of the potential morphospace of diverse flowering-plant clades when sampling of individual parts is unequal; (ii) by circumventing asynchronicity in anthesis of different floral parts; and (iii) by incorporating variation in copy number of parts within structures. This approach, for which the code is available as Supplemental Material, can be used for any flower as well as numerous plant structures and can be used to form an appropriate basis for future geometric morphometric and related studies starting from 2D pictures.

## ACKNOWLEDGEMENTS

Many thanks to Walter den Hollander, Reinout Havinga and Carlien Blok from the Hortus Botanicus Amsterdam for taking in and maintaining the *Pelargonium* collection. In addition, we thank Melanie Wiethölter and Kai Müller from Münster Botanical Garden, Martin Smit and Peter Jansen from Stellenbosch Botanical Garden, Anthony Hitchcock and Roger Oliver from Kirstenbosch Botanical Garden (Cape Town), and Werner Voigt and Ricardo Riddles from Karoo Desert National Botanical Garden (Worcester) in South Africa for granting me access to their collections. Luuts and Riet Feenstra, thank you so much for welcoming us in your home and opening your collection so many times.

Also the Family Mijnbergen, Arjen de Graaf, and Esther van der Velde, thank you for having us. In addition, we thank all other members of the Nederlandse Pelargonium Vereniging for their kind interest. Liesbeth Sluiter, Ronald Bakker, Ronald Flipphi, Rolf Roos, Paul Raadschelders, thank you for the amazing trip. We thank Nynke Groendijk-Wilders for her help with the digitalisation process.

### Funding
The authors received no funding for this work.

### Competing Interests
The authors declare that they have no competing interests.

### Author Contributions
- Sara J. van de Kerke developed the code, performed the experiments, analysed the data, prepared figures and/or tables, authored or reviewed drafts of the paper, and approved the final draft.
- Tiemen van Engelenhoven performed the experiments, authored or reviewed drafts of the paper, and approved the final draft.
- Anne L. van Es performed the experiments, authored or reviewed drafts of the paper, and approved the final draft.
- Laura Schat performed the experiments, authored or reviewed drafts of the paper, and approved the final draft.
- Lisa M. van Son performed the experiments, authored or reviewed drafts of the paper, and approved the final draft.
- Sverre Vink performed the experiments, authored or reviewed drafts of the paper, and approved the final draft.
- Lia Hemerik developed the code, authored or reviewed drafts of the paper, and approved the final draft.
- Robin van Velzen developed the code, authored or reviewed drafts of the paper, and approved the final draft.
- M. Eric Schranz conceived and designed the experiments, authored or reviewed drafts of the paper, and approved the final draft.
- Freek T. Bakker conceived and designed the experiments, authored or reviewed drafts of the paper, and approved the final draft.

### Data Availability
Data is available at 4TU.centre:

van de Kerke, S.J. (Sara) (2019) Pelargonium geometric morphometric data. 4TU. Centre for Research Data. Dataset. DOI 10.4121/uuid:7457eadf-5ad6-4221-b4ea-85c73b91098b.

## Supplemental Information

Supplemental information for this article can be found online at http://dx.doi.org/10.7717/peerj.8823#supplemental-information.

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
