# Peer review of "Capturing variation in floral shape: a virtual3D based morphospace for Pelargonium"

_PeerJ, doi:10.7717/peerj.8823_

## Round 0.1 · original submission · Major Revisions

The paper attracted two conflicting reviews with reviewer 2 accepting the paper with minor revision and reviewer 1 rejecting it. Both reviews are comprehensive and offer many constructive criticisms.

Reviewer 2, despite rejecting the paper, was quite enthusiastic about it. The major criticism was articulated thus: "I don’t understand the value of the exercise if the 3D virtual flowers don’t reflect real allometric constraints observed among individual flowers. How can conclusions regarding the biology of the relationship between tube length and petal size in this group be drawn from these studies". If this is the case, it throws doubt on the biological relevance of the method.

The authors will need to provide in a major revision a convincing argument that counters this criticism if this paper is to be reconsidered for publication.

·

Basic reporting

English language: This manuscript is mix of grammatically correct, easy to read sentences and sentences that are more difficult to understand.

For example:

Line 56: “The total variation in shape of a clade….” Do clades have shapes?

Line 59: (GMM) in parentheses should follow the first use of the phrase in line 58, and then be used without parentheses in line 59.

Line 65; “…used here as a proxy to determine inclusiveness or accuracy” is unclear. In fact, the entire paragraph could be rewritten to increase readability.

Line 79: This sentence might be clearer if it read: However, studies have shown that different organs within individual flowers are not synchronized in their development…”

Into and Background show context:

1. Lines 103-107: I thought (and the floral morphologist Peter Endress, who reviewed our paper, agreed) that we (Tsai, et al. 2018) clearly showed that the “nectar spur” is in fact, NOT A SPUR adnate to the pedicle, but instead is a unique type of nectar tube that forms via elongation of the receptacle. Citing our paper amongst these others that assume (without doing and development) that the spur is adnate to corolla is misrepresenting our work. If the authors disagree with our interpretation, then it seems to me that good science dictates that they justify calling the organ a SPUR. This may seem like a trivial point to a non-morphologist, but it is important to morphologists (and it matters with respect to gene ontologies). A relatively easy work around is simply to replace “spur” everywhere it occurs, including the figures and legends, with “tube” or “Nectar Tube.”

To that end, the pedicle is not “occupied” or “filled” by the nectar tube. Instead, the comparison is between pedicle + receptacle length relative to receptacle length.

2. An issue that should be considered in the introduction is why describing a morphospace is important. There does not seem to be a central research question, e.g. line 91.

Line 116-117: Why/how is the short nectar tube of P. cotyledonis evidence that the pollination syndrome is highly specialized? Generally more actinomorphic corollas with short nectar spurs indicates greater generalization.

Literature well referenced and relevant:

Line 79: (refs)????
See above re. Tsai et al. 2018.
Line 101, van de Kerke et al. 2019 show a “pattern of deep splits as well as more recent species radiations.” I don’t think the additional reference to Bakker, et al. 2005 is necessary.

Figures relevant, high quality:

Fig. 2. Show images with larger points for sliding landmarks. The small points are not visible if the figure is printed on a standard B&W printer.

Raw data supplied: NO

Experimental design

For the most part, the experimental design is clearly described, with some exceptions.

Line 147: “We selected flowers with corollas and flowers with stamens, used as proxy for full anthesis of the spurs.”
Does this mean that the authors removed the petals in order to photograph flowers from the side? Or that they waited until the petals fell off. If this is the case, then the anthers would have been past functioning since most species are protandrous.
What do anthers have to do with “full anthesis of the spurs”? Are you using anthesis to mean fully elongated or fully mature?

Line 153: A description of how primary landmarks were chosen should be added as an appendix, and it would be good to provide more description of what a sliding landmark is, and how they were determined.

Line 161: Clarification is needed regarding “midrib, primary and secondary veins…as a guide.” How was the midrib vein identified? I don’t see it in this image.

**Most important point** I am confused as to why the authors did not photograph a flower from the top, and then remove the petals, and photograph the SAME FLOWER from the side at the same time? I don’t understand what is meant by Line 173: “(as a result of the separate sampling in order to avoid of asynchronization).” Given incredibly high variation in floral size and color within and among individuals of a species, mixing and matching flowers of varying petal dimensions with random nectar tube dimensions from another flower, the authors are creating virtual flowers that may not exist. Within a species, shorter nectar tubes could be associated with smaller petals, while longer tubes are associated with larger petals. If this is the case, the size-free shapes would be very similar, and the degree of variation among virtual 3D flowers within a species would be much lower. If tube lengths are independent of petal sizes, then the mix and match approach here is representative of true variation.

Validity of the findings

1. The descriptions of morphometric variation among nectar tubes and among petals are interesting and relevant to understanding overall floral variation in this genus.

2. Unfortunately, while the method of creating the 3D virtual flowers is very cool, I don’t understand the value of the exercise if the 3D virtual flowers don’t reflect real allometric constraints observed among individual flowers. How can conclusions regarding the biology of the relationship between tube length and petal size in this group be drawn from these studies.

3. By adding size back into the analysis after the procrustean alignment, I assume the authors are describing multivariate variation in size and shape with the PCA analysis, not just shape. This point should be addressed more explicitly.

Additional comments

I appreciate that this study involved a lot of work, and I wish I wasn’t so skeptical of the linking of two different data sets. Even though reviewers are not supposed to offer opinions, I can see a couple of ways forward. One would be to measure another limited set of flowers (say 10 per species on different plants) to demonstrate how consistently petal size and nectar tube size scale among flowers on maybe 10 species (not the full morphometric data set, just a measure of petal size and nectar tube length); use one "homologous" time point. If there are tight allometric constraints, use these to define your parameters for mixing and matching. An additional approach is to not apply the 3D approach to these data and instead analyze each data set separately, both with size included, and size-free, i.e. how much floral variation among and within species is due to differences in floral size versus shape at anthesis?

Reviewer 2 ·

Basic reporting

Comments regarding this section were made throughout the annotated manuscript I am joining to this review.

Experimental design

Comments regarding this section were made throughout the annotated manuscript I am joining to this review.

Validity of the findings

Comments regarding this section were made throughout the annotated manuscript I am joining to this review.

Additional comments

This manuscript focuses on an interesting topic that deals with the usefulness of building a composite 3D dataset from a set of two 2D images for more representative geometric morphometrics of floral shape variation. In general, the manuscript is short, appropriately illustrated and straightforward to read. The hypotheses are clearly presented and the overall methods seem appropriate for the study.

I am enclosing a file containing all detailed comments as well as suggestions and corrections throughout the manuscript.

- Regarding the analyses, a section discussing ways to assess and take into account measurement error due to imaging (i.e. positioning the specimen before taking pictures) and digitizing (i.e. precision of acquiring landmarks) is missing here. Procrustes ANOVAs and ANOVA using centroid size can be used to assess the magnitude of measurement error due to imaging and digitizing versus the biological component of interest (e.g. variation among flowers or differences between left and right petals). This is an important step in GMM studies to ensure that the measured variation does not contain or contains a negligible amount of error. See for instance Savriama (2018) for a recent account on how to assess measurement error in floral shape

Savriama, Y. (2018). A Step-by-Step Guide For Geometric Morphometrics Of Floral Symmetry. Frontiers in plant science, 9, 1433.

- How is symmetry of the corolla taken into account here? According to the visualizations in Figure 4D and S4, the mean shapes are not symmetric indicating that zygomorphy was not taken into account at least with the method of bilateral object symmetry (Mardia et al. 2000, Savriama & Klingenberg 2011).
I had a look at the corresponding R script and there is no mention of such analysis either.
Given that symmetry is present in the studied species and that according to l.110-111: “ranges from highly zygomorphic (P. fulgidum) to almost actinomorphic (P. cotyledonis).”, I would have expected such important feature of floral shape to be taken into account here.
I would be very interested in seeing such analysis included in this manuscript specifically since the methods associated with bilateral object symmetry unambiguously separate a component of symmetric variation (i.e. variation among flowers) and the asymmetry (i.e. differences between left and right petals). For instance, a visually suspected actinomorphic variation in these flowers could be accurately detected and quantified using these approaches.
If asymmetry is not of interest then analyses can be carried solely on the component of symmetric variation which also further reduces the dimensionality of the data (Mardia et al. 2000, Berger et al. 2017). This additional point would be useful here given the high number of landmarks and semilandmarks used in the present work.

Berger, B. A., Ricigliano, V. A., Savriama, Y., Lim, A., Thompson, V., & Howarth, D. G. (2017). Geometric morphometrics reveals shifts in flower shape symmetry and size following gene knockdown of CYCLOIDEA and ANTHOCYANIDIN SYNTHASE. BMC plant biology, 17(1), 205.

Mardia, K. V., Bookstein, F. L., & Moreton, I. J. (2000). Statistical assessment of bilateral symmetry of shapes. Biometrika, 285-300.

Savriama, Y., & Klingenberg, C. P. (2011). Beyond bilateral symmetry: geometric morphometric methods for any type of symmetry. BMC evolutionary biology, 11(1), 280.

In my opinion, the manuscript is suitable for publication after these corrections as well as other points raised in the annotated manuscript I am joining to this review have been addressed.

Best wishes.

Annotated reviews are not available for download in order to protect the identity of reviewers who chose to remain anonymous.

---

## Round 0.2 · accepted · Accept

It is clear to me that you and your team have put a great deal of effort into revising this manuscript. In doing so, you have responded convincingly to the substantive criticisms made by Reviewer 1.